# Effect of Hydrogen Bonding on the Surface Tension Properties of Binary Mixture (Acetone-Water) by Raman Spectroscopy

**Nannan Wu [1,2], Xin Li [1], Shiliang Liu [3], Mingzhe Zhang [1] and Shunli Ouyang [1,*]** 

[1]   Key Laboratory of Integrated Exploitation of Bayan Obo Multi-Metal Resources, Inner Mongolia University of Science and Technology, Baotou 014010, China; woshinannan04@imust.cn (N.W.); leexinchc@gmail.com (X.L.); imust2016023081@163.com (M.Z.)

[2]   College of Science, Inner Mongolia University of Science and Technology, Baotou 014010, China

[3]   School of Energy and Environment, Inner Mongolia University of Science and Technology, Baotou 014010, China; 13224726207@163.com

[*]   Correspondence: ouyangshunli@imust.cn; Tel.: +86-138-4726-7509

**Abstract:** The structure and properties of water and aqueous solutions have always been the focus of attention. The surface tension of acetone aqueous solutions were measured by using Raman spectra in different molecule environments, and the changes of surface tension were analyzed with hydrogen bonding in the mixtures. In this case, OH stretching bands were fitted into three Gaussian components and then assigned to different hydrogen-bonded structures. Furthermore, it can be concluded that the changes of microstructure on the binary mixture solutions exhibited regularly with the additive acetone, indicating that there showed the correlation between surface tension and relative peak intensity. The results show that the strengthening of hydrogen bonding between acetone and water will gradually weaken the surface tension of the solutions. It was confirmed that there showed the relationship between the microstructure and macroscopic properties of the aqueous solutions by the basis experiment data using a spectroscopy method.

**Keywords:** raman spectroscopy; binary solutions; surface tension; hydrogen bond; acetone

## 1. Introduction

Water, as a simple and peculiar substance in nature, was widely used in many fields and has become a hot science topic that has been received extensive attention [1–5]. It is playing a vital role in daily life, industrial production, and scientific research. For natural creatures, all life-related energy exchanging the chemical reactions are carried out in a liquid environment. Despite the water possess a simple structure, it exhibits the complex properties. Hence, it is familiar and unfamiliar with the common substance of water. So far, water is the only substance been found with abnormal density, and it expands in the process of liquid-solid phase transformation [5], reaching the maximum at 276.984 K, and density of other substances reaches the maximum at the solid state. Russo et al. [6] found that the physical properties of water near 276.984 K were related to the transformation of different hydrogen bonding modes in water by means of molecular mechanics simulation. More recently, the mysterious side of the water structure and interaction have been attempted to discuss and explore in the aqueous mixtures [7,8]. It was found that the molecular structure of water changed during in the process of phase transition due to the variation of status on hydrogen bonding, transformed from chain, cyclic annular structure to a stable three-dimensional network structure. It was similar to ice. At present, although there are many debates on liquid water [9–11], a high consensus and reasonable structural model, the tetrahedral complex structure, being similar to ice, exhibits the dynamic variation between

continuous reorganization and deconstruction at an averaged status [11–15]. The effects of structure of aqueous solutions and the hydrogen bonding effects on structure from different perspectives were investigated through different experiment analysis [9,13,16].

Hydrogen bonding, as the basis of water structure, is an important microscopic effect in aqueous systems [17–19]. As a bridge between bonding molecules, it affects the microscopic heterogeneity and microstructure of the solutions. The transformation of hydrogen bond structure led to abnormal changes of other properties in aqueous solutions, resulting from the change of microstructure of water owing to hydrogen bonding networks. It was found that liquid molecules with a hydrogen bond structure are to be associated with each other, resulting in the increase of a solution density. As before, the influence of hydrogen bonding on dielectric constant and boiling point of several kinds of solutions with hydrogen bonding and non-hydrogen bonding were investigated [20]. The abnormal dielectric constant exhibited for the hydrogen bonding system. Meanwhile, the boiling point of substance was enhanced due to the hydrogen bond, been reached up to 76 °C. It was similar to the other research [21]. The dissolution and solubility of the solute were improved by hydrogen bonding in the aqueous solutions. In our previous research, the mixture solutions showed a lower freezing point than the solute due to the hydrogen bonding of solute and water, thus having a better antifreeze effect [22]. Afterwards, Wang et al. [23] found that hydrogen bonding changes the molecular structure of the azole explosives and increases the melting point. The melting point could be reduced by weakening the hydrogen bonding with adding the suitable substituent. Moreover, the hydrogen bonding between polymer chains and water molecules was determined by the viscosity in polyvinyl alcohol aqueous solutions [24]. Undoubtedly, the rheological properties of the solutions depended on the relative strength of hydrogen bond. Hence, the hydrogen bond structure will be broken if the temperature, pressure and degree of hydrolysis increased or decreased the viscosity. Thus, the hydrogen bonding network, as a microscopic interaction, was related to water molecular structure, being a key factor for many anomalous properties of water with the change of microstructure [1,25–27]. Furthermore, it was confirmed that the microscopic intermolecular interaction of hydrogen bonding has a significant effect on macroscopic properties.

Nowadays, several kinds of methods containing using the theoretical calculations, simulations and modern experimental detection, were used to investigate water properties and structure. An important branch of modern spectroscopy, Raman spectroscopy, is widely used, which has many advantages, for example, non-destructive, non-contact, in-situ testing, etc. It can avoid direct interaction with molecules and molecular structure, especially useful for of structure and weak interactions of aqueous systems, which was different from other high-energy ion testing. It was a better way to understand aqueous solutions [22,28,29]. So far, despite the fact that a lot of work has been done on aqueous solutions by Raman spectroscopy, the diversity of the situations investigated has led to various interpretations and/or controversial discussions on the structure of water. [27–31]. The important breakthrough of water structure was proposed by the Raman vibration peak of OH-bond, as a signal region [22,28,30,32–34]. Gaussian fitting, an advantageous method in peak fitting, was used to investigate the vibration modes of OH bond. As for the OH bond, Raman peak was divided into two components; including the symmetric stretching vibration in low wavenumbers and the antisymmetric stretching vibration [21]. Walrafen et al. [34] described the Raman peaks of OH bonds with diverse wavenumbers while $H_2O$ was in different hydrogen bonding environments. OH stretching band was often fitted with different Gaussian components (five [35–37], four [38,39], three [40], two [41,42]) and each sub-band was further attributed to different structure models. In order to explore the structure and the properties of aqueous solutions, an attemptwhich is relevant to the present effort is devoted to analyze the typical hydrogen bond system solutions of acetone-water on the relationship between hydrogen bonding and solutions structure and on the surface tension with increasing amount of acetone.

## 2. Experiment

### 2.1. Samples and Preparation

Ultrapure water, HPLC grade, with resistivity of 18.2 MΩ cm purchased from J&K Scientific LED, (Beijing, China). Acetone used in the experiment is in purity greater than 99.9%, without further purification before experiment. Acetone-water mixtures were formulated at the molar ratios ($nCH_3COCH_3:nH_2O$) as 1:20, 1:10, 1:6, 1:5, 1:4, 1:3, 1:2, 3:4, 1:1, 3:2, 2:1, 5:2, 3:1, 4:1, 5:1, 6:1, 7:1, 8:1, 9:1, 10:1, 11:1, 12:1, 13:1, 15:1, 16:1, 18:1, 19:1 and 20:1.

### 2.2. Experiment Apparatus

The surface tensions were measured by ZL-10 type automatic digital display interface tension meter purchased at Zibo Aiji Electric Co., Ltd. (Zibo, China), with the measuring range at 2~200 mN/m, accuracy and resolution respectively are 0.1 mN/m and 0.2 mN/m. In the experiment, the average of surface tension were geted with the multiple times recorded surface tension of pure water, pure acetone, and the mixtures respectively. The surface tension of pure acetone and water were 24.5 mN/m and 72 mN/m, respectively.

Plural of spectrum were collected by the Raman spectrum system constructed independently by our laboratory equipped with Andor Sham-rock SR-500i-C-R type Raman spectrometer and Andor iDus series water-air cooled CCD (Charge Coupled Device) detector product by UK ANDOR company, and a 1200 grove/mm grating with a wavelength resolution about 0.05 nm. In the experiment, 50 times long focus Olympus objective lens (parameter: $50\times/0.35$) is used to focused on samples, adjusting the three-dimensional sample stage height to complete the focal point, focusing image collected by SunTime130E CMOS color digital camera, excited at the 532 nm line of semiconductor laser which output power is 25 mW. And then the Raman spectrometer was calibrated with a diamond standard sample. All the signal data were gathered at room temperature with 5 s exposure time, 2 times accumulation number, 5 s accumulation cycle time, and the scanning range is 0 to 4000 cm$^{-1}$.

## 3. Discussion

The relationship between the surface tension and the mole ratio were shown in Figure 1. The surface tension of the solutions decreases significantly with the increase of amount of acetone. Interestingly, the reduction of surface tension becomes obvious until the molar ratio is about 1:1. However, the surface tension decreases gentlelyor the gradient of tension changes is small as the molar ratio is greater than 1:1. It was found that the changes on surface tension of the solutions were affected by two pure solutions, which are directly conductive to the content of it. As shown in Figure 1, the two pure solutions-dominated regions exhibitwhile it is near the two pure solutions. Meanwhile, the surface tension of the solutions does not vary linearly with the molar ratios of the two solutions, and there showed the great difference of trend at the scale of molar ratio. It can be concluded that the change of surface tension is affected by the microscopic interaction. Otherwise, maybe it was limited by the changes of microstructure of mixture solutions.

Figures 2 and 3 show the Raman spectra of C=O bond and OH bond stretching vibration in different molar ratio acetone aqueous solutions at normal temperature, respectively. In order to understand the interaction and microstructure changes in the mixed solutions, the Raman spectra of different solutions were measured, and the trend of the Raman peaks of the C=O stretching bond and the OH bond in solutions with different molar ratios was obtained. Apparently with the addition of acetone, the stretching vibration peak of the C=O bond moves toward the high wavenumber. It is confirmed that hydrogen bond is formed between the O atom in acetone and the H atom of OH in $H_2O$.

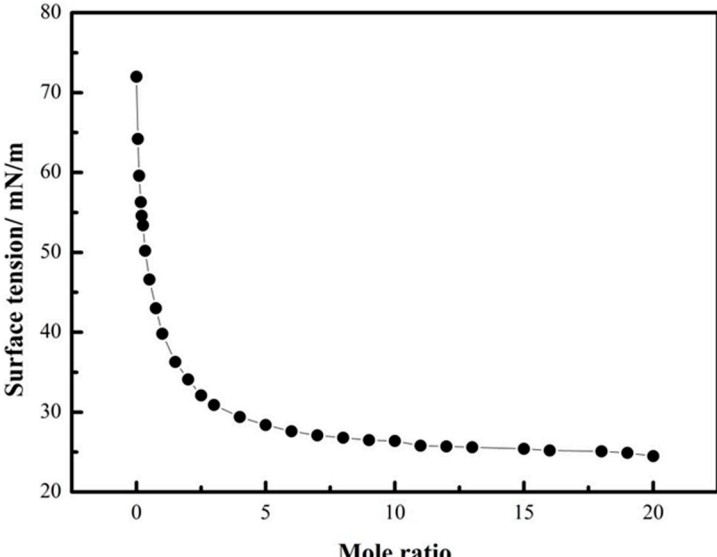

**Figure 1.** The relationship between surface tension and mole ratio of acetone aqueous solutions.

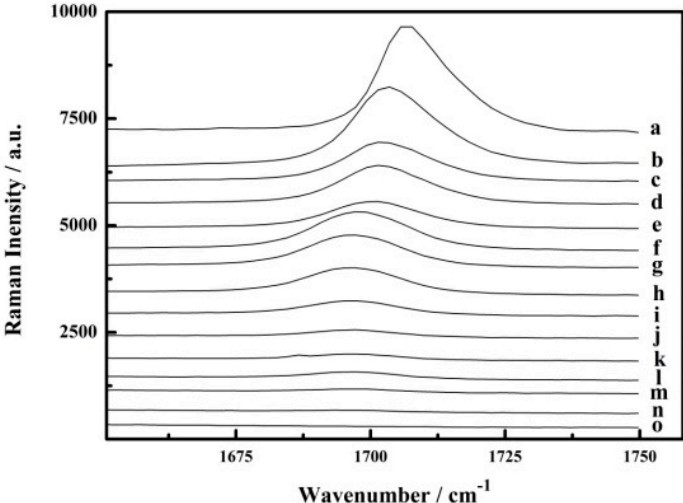

**Figure 2.** Raman spectra for C=O stretching bond of pure acetone and acetone aqueous solutions with different moral ratios ($nCH_3COCH_3:nH_2O$). a to o respectively correspond to pure acetone, 20:1, 10:1, 9:1, 5:1, 3:1, 3:2, 1:1, 1:2, 1:3, 1:4, 1:5, 1:10, 1:20 and water as a comparison.

In the range of low concentration which is less than 1:1 for the molar ratio, the vibration peak moves toward a high wavenumber with the addition of acetone, indicating that the number of hydrogen bonds in the solutions increases and the hydrogen bonding is strengthened, which promotes the long elongation of the C=O bond. At the same time, the vibration force constant becomes small. The molecular structure in pure water is divided into three categories: free water molecule, partially hydrogen-bonded water structure in which hydrogen bond with one or more surrounding molecules was formed, and a tetrahedral hydrogen-bonded structure in which hydrogen bonding with one and three surrounding water molecules are connected [28,29]. The vibrational band of the OH bond is considered as the characteristic band of $H_2O$. As shown in Figure 3, the shoulder peak near 3240 cm$^{-1}$ is identified as the Raman spectrum peak of the fully hydrogen-bonded tetrahedral structure water in the solutions, and the peaks around 3420 cm$^{-1}$ and 3550 cm$^{-1}$ are assigned to the Raman peak of the partially hydrogen-bonded structure water. It was revealed that the peak intensity of the fully hydrogen-bonded gradually weakened with the increasing content of acetone in the solutions system. Nevertheless, there was hardly changed in the shift of peak position significantly. In addition, the peak of partially hydrogen-bonded water moves toward high wavenumbers, and the relative peak intensity

of the component increase gradually. As a result, due to the content of acetone increases, the hydrogen bonding between acetone and water enforce the broken of fully hydrogen-bonded structure, and the amount of partially hydrogen-bonded structures increase. Besides, the addition of acetone will reduce the polarity of water. The electron cloud of OH bond in the water was toward to the O atom originally and then moved to the side of the H atom. The electron cloud density deviates from the center of the two atoms, resulting in the reduction of vibration force constant and the increase of vibration frequency. Hence, the red-shift occurred in the OH bond. As so, the electrons in H were attracted to the oxygen side of the acetone, the O atoms in the water moved toward the side of H because of the H atoms to be attracted by hydrogen bonding, so that the OH bond length is reduced to achieve the electron density balance.

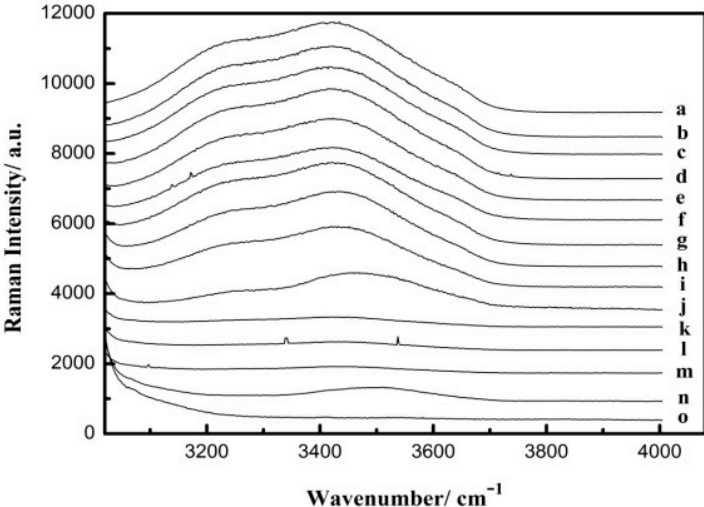

**Figure 3.** Raman spectra for OH stretching bond of water and acetone aqueous solutions with different moral ratios ($nCH_3COCH_3:nH_2O$). a to o respectively correspond to pure water, 1:20, 1:10, 1:5, 1:4, 1:3, 1:2, 1:1, 3:2, 3:1, 5:1, 9:1, 10:1, 20:1 and pure acetone as a comparison.

In order to explore the hydrogen bonding between two kinds of molecules, the Raman shifts of C=O bond and OH bond with the variation of molar ratio were introduced, as shown in Figures 4 and 5. The Raman shift of C=O bond in different molar ratio acetone aqueous solutions were shown in Figure 4. With the increasing content of water, the characteristic peak of C=O stretching bond shifts towards the low wavenumber gradually and the maximum shift was about $11cm^{-1}$. However, the Raman shifts of the OH stretching bond moved to a high wavenumber with the acetone addition in the solutions, reaching the maximum shifts, $88 cm^{-1}$, as shown in Figure 5. Compared to Figures 4 and 5, the blue shift of the OH bond is more significant than the C=O bond. The Raman shift of OH bond is about 8 times of C=O, which indicating that there are great influence on the water structure with the addition of acetone. It is resulted in the overall structure of water in the solutions and the average number of hydrogen bonds in the bulk water reduced. We have found from the hydrogen bond study of Tetrahydrofuran-water mixture that with the addition of water [43], the C-H bond length decreases gradually and shows a concentration dependence in the Raman spectrum. It should be noted that there existed similar changes in the acetone water binary solution system. With the increase of acetone in the binary solution, the combination of acetone and water molecules converted from single acetone molecule which acted on multiple water molecules, to single acetone molecule with the combination of single water molecule. The C=O bond length decreases as the concentration increases, and the frequency amplitude decreases in the Raman shift.

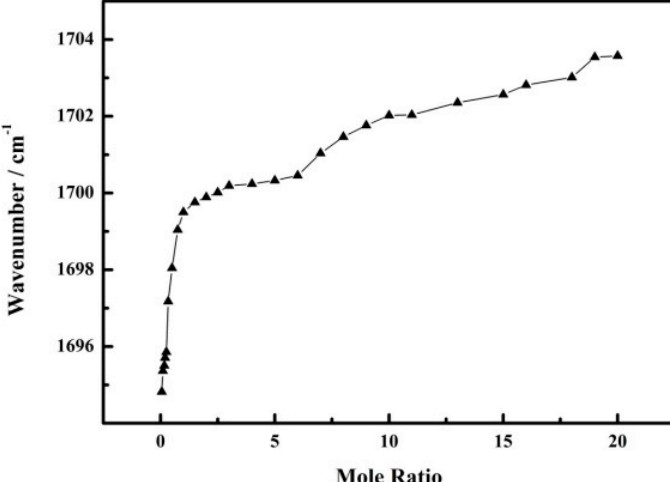

**Figure 4.** Raman shift of C=O stretching bond in acetone aqueous solutions at different mole ratios.

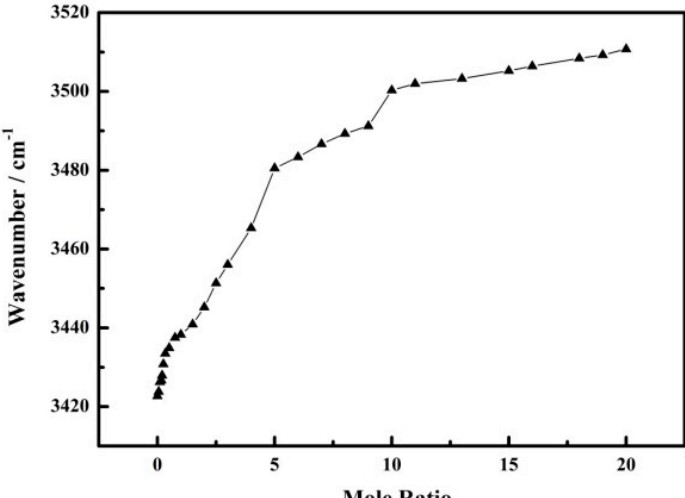

**Figure 5.** Raman shift of OH stretching bond in acetone aqueous solutions at different mole ratios.

There showed more advantages for acetone molecules obtained water molecules from the tetrahedral hydrogen-bonded structure than the water molecules in the process of hydrogen bonds formed in the solutions with the hydrogen bonds been produced. The stable tetrahedral hydrogen-bonded network structure in the bulk water was broken so that some water molecules were forced to detach from the network. The water molecules formed a strong hydrogen bond with acetone, and increased the content of partially hydrogen-bonded water in the bulk water. It led to drop the surface tension rapidly in the water-rich region, as shown in Figure 1. As shown in Figure 4, as the content of water increased, red-shifts of the C=O bond in acetone exhibited, and the Raman shift rate gradually increased with the change of molar ratio, from a high molar ratio to a low because of the hydrogen bonding interaction between acetone and water. The hydrogen bonds increased with the increasing water.

It can be seen from Figures 4 and 5 that the Raman shifts of the OH bond and the C=O bond has a similar tendency with the change of molar ratio. In the water-rich region, the two characteristic peaks moved fast towards high wavenumber and possess a large amplitude, so did not in the acetone-rich zone. This is because the hydrogen bonding in the mixed solutions is strengthened with addition of acetone, and it changed the interaction between molecules in the solutions. As shown in Figure 1, the surface tension changes sharply with the additive acetone. In the acetone-rich region, the interaction between acetone and water is weakened due to the decrease of water molecules, leading to the simple

molecular structure in the solutions. The solution properties were mainly determined by acetone. The variation of surface tension can be expressed as a long platform, which was consistent with the previous discussion. Thus, the acetone had great influence on the molecular structure and microcosmic effect in the binary system of rich water area.

The effect of the acetone on the microstructure and the transformation of hydrogen bonding patterns using the Raman peak of OH bond on the water structure been fitted into several sub-components were investigated in various structural models of aqueous solutions. Accordingly, OH bonds are often divided into multiple sub-peaks. Risovic et al. divided the OH bond into two sub-bands at 3424.8 cm$^{-1}$ and 3206 cm$^{-1}$ been attributable to non-H-bonded and H-bonded species, respectively. Three sub-peaks of OH bond were divided and located at 3241 cm$^{-1}$ for the ordered water structure (ice-like structure), 3461 cm$^{-1}$, 3655 cm$^{-1}$, respectively, which was consistent with the disordered structures (defective structures) [40]. Moreover, Li et al. [37] fitted the OH bond into five Gaussian components, which was assigned to fully or partly hydrogen bonded water structure. As so, the OH stretching band was roughly fitted into three sub-peaks, being located at 3245 cm$^{-1}$, 3550 cm$^{-1}$, 3420 cm$^{-1}$, respectively, namely PekI, PekII, PekIII. The PekIwas assigned to the fully hydrogen-bonded tetrahedral water and PekII as well as PekIII were assigned to the partially hydrogen-bonded water. The peak intensity of each peak obtained was noted as $I_I$, $I_{II}$, $I_{III}$, and the relative abundance of PekI was calculated as $I_I/(I_{II} + I_{III})$.

Figure 6 shows the relative abundance of the fully hydrogen-bonded tetrahedral structure and partially hydrogen-bonded structure in the mixture with different mole ratios. As the molar ratio increased, the relative peak intensity of PekI decreased gradually, meaning the change of the composition of the fully hydrogen-bonded structure and partially hydrogen-bonded structure in the solutions. The fully hydrogen-bonded structure is broken gradually and converted to partially hydrogen-bonded water with the acetone addition. The original stable microstructure in the aqueous solutions was broken and the water molecules and acetone molecules formed a new type molecular structure through hydrogen bonding. As the content of acetone rise continuously, the molecular structure in the binary system changes gradually. Importantly, the total hydrogen bond water decreased gradually, and the relative content of hydrogen bond water reached a level of stability. Furthermore, a plurality of water molecules been surrounded by acetone molecules form a cluster structure with the increasing molar ratio, which was non-uniform microscopically and uniform macroscopically. The interaction between acetone and water can be expressed by the change of the relative content of two hydrogen bond structures. It is agreement with the change of surface tension with the molar ratio very well. It can be confirmed that the change of surface tension were associated with intermolecular interactions in solutions systems.

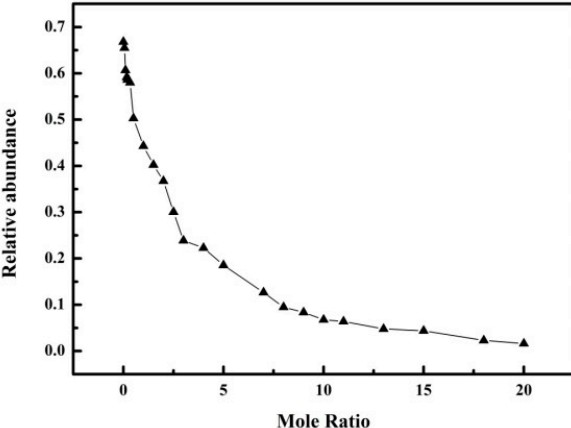

**Figure 6.** Relative abundance of OH sub-bond in acetone aqueous solutions with different mole ratios.

To further understand the effect of the change of molecular structure on surface tension in solutions, a graph of the relative abundance of tetrahedral hydrogen bonds structure and surface tension was obtained, as shown in Figure 7. And the trend of nonlinear variation was fitted and the correlation coefficient $R^2$ was 0.98. The results showed that there was a good exponential relationship between surface tension and relative abundance. The surface tension of solutions is dominated by acetone solutions when the relative abundance was in the low region with low water-content and few tetrahedral hydrogen bonds. With the increasing water-content and strengthen of hydrogen bonding between water and acetone molecules, the molecular structure will be changed. Meanwhile, the tetrahedral hydrogen-bonded structure and the surface tension increased. The hydrogen bond between acetone and water is no longer more obvious than that of tetrahedral hydrogen bond in water as the water-content and the tetrahedral hydrogen bond increased continuously in a zone of relative high abundance. Thus, the acetone exhibited the characteristics of non-homogeneity on the micro level and homogeneity on the macro level in water solutions. Water molecules had great influence on the structure of solutions. The molecular structure becomes stable so that the surface tension increases rapidly in this given zone. Afterwards, the water-content increase continuously, the change of surface tension end up at the pure water surface tension. It was consistent with the previous research, which was similar to the results of acetone with the additive water. It showed that hydrogen-bond between acetone and water would like affect the surface tension in this binary system. And acetone played an important role in the process of change of surface tension.

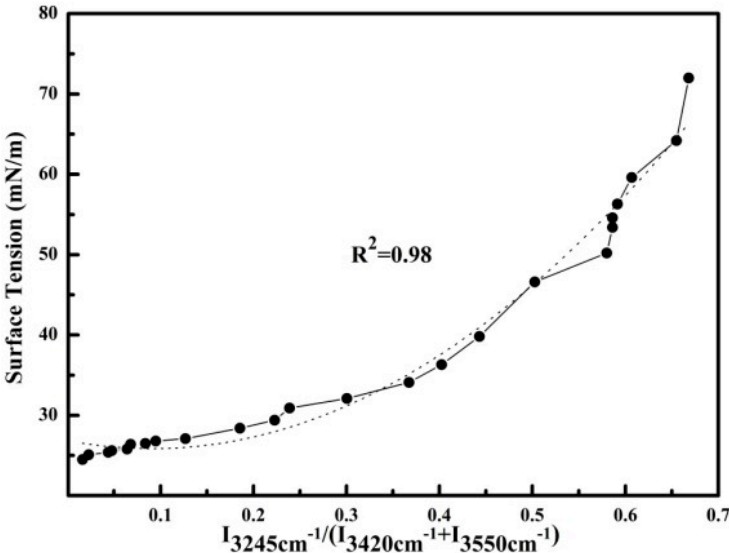

**Figure 7.** Relationship between surface tension and relative abundance of tetrahedral hydrogen bonds.

## 4. Conclusions

Raman spectra and surface tension of acetone aqueous solutions with different mole ratio were measured. The variation of surface tension with molar ratio of mixtures, the spectral characteristics of C=O bonds and OH bonds containing hydrogen bonding was obtained. The Raman spectra of OH bond were fitted into three Gaussian components and assigned to two Hydrogen-bonded environments. The results showed that the change of microstructure in the binary mixture were caused by the hydrogen bonding interactions. The trend of fully hydrogen-bonded tetrahedral structure toward the partially hydrogen-bonded structure of water occurred with additive acetone. The hydrogen bonding between acetone and water affected the change of surface tension significantly. The weakening of surface tension was related to the enhancement of hydrogen bonding. A novel way for establishing the relationship between the microstructure and macroscopic properties of solutions was provided, which offered the basic data for the experimental analysis of spectroscopy in water science.

**Author Contributions:** Conceptualization, S.Y.; software and investigation, N.W. and S.L.; resources, N.W. and S.Y.; writing—original draft preparation, N.W., X.L. and M.Z.; funding acquisition, N.W. and S.Y.

**Funding:** This research was funded National Natural Science Foundation of China (Grant Nos. 21864019, 11364027, 21363013, 11564031), Natural Science Foundation of Inner Mongolia Autonomous Region (Grant No. 2018LH02004), Program for Young Talents of Science and Technology in Universities of Inner Mongolia Autonomous Region, China (Grant No. NJYT-17-B10) and the Special Foundation of Instrument Research and Development in Inner Mongolia University of Science and Technology, China (Grant No. 2015KYYQ06).

**Conflicts of Interest:** The authors declare no conflict of interest.

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
