# Peer review of "Effect of Hydrogen Bonding on the Surface Tension Properties of Binary Mixture (Acetone-Water) by Raman Spectroscopy"

_applsci, doi:10.3390/app9061235_

Round 1
Reviewer 1 Report
Throughout the paper- Always open one space between numbers and the respective unit symbol
Thorough revision of used English language. Examples:
Abstract: The structure and properties of water and aqueous solution…./ The structure and properties of water and aqueous solutions….
29-.. We are both
R: Substitute this verbal form.
32-… The density ….
R: Its density…
107- Raman spectra…
R: Plural of spectrum
114- …. is ~ 25 mW. And then the Raman spectrometer was….
R: Revise punctuation
116- All the signal dates were
R: data?
119- ….the surface tension respectively measured multiple times of pure water…
R: the multiple times measured surface tension of pure water ……mixtures respectively.
122- R: acetone and water at 24.5mN/m and 72mN/m respectively.
125- …significantly. And
R: See above
279-280-This study provides a novel way for establishing the association between the microstructure and macroscopic properties of solution, and provides (R: repeated/choose another word) basic (?) data for the spectroscopy experimental study
Author Response
Response to Reviewer 1 Comments
Dear Reviewer,
Thank you for your comments concerning our manuscript (ID: applsci-431965). and giving us the opportunity to modify the paper. Those comments are all valuable and very helpful for revising and improving our paper. We have studied comments carefully and have made correction which we hope meet with approval. Revised portion are marked in blue in the paper. The main corrections in the paper and the responds are as flowing:
Point 1: Throughout the paper- Always open one space between numbers and the respective unit symbol.
√Dear editor and reviewers,
We have added a space between all numbers and units. And all modifications are marked in blue.
Please check it in the manuscript.
Point 2: Thorough revision of used English language
√Dear editor and reviewers,
We have made a substantial revision of the English language in the text, and the modified parts are marked in blue.
Please check it in the manuscript.
We are look forward to hearing from you.
Best regards,
Shunli Ouyang
Corresponding authors

Reviewer 2 Report
The paper on " Investigation on surface tension properties of binary solution (acetone-water) based on hydrogen bonding by Raman spectroscopic " , is an important contribution. However, the English has to be improved. Eg., inclusion of either " Raman spectroscopy " or "Raman spectroscopic methods " in the title would be more suitable. There are a lot of places where English has to be improved.
Some repetition of texts are present, eg., lines 128-131.
Figure 2,3 or 6 captions contain "acetonitrile ". But this study is based on water and acetone mixtures. Please correct it.
However, in the discussion section, lines 164-166 , the authors have argued that the combination of acetone and water reduces the polarity of water and thus OH bond gets elongated and there is a blue shift in Raman spectra. But this is contradictory to Figure 5. As I understood, Figure 5 shows that with increase in acetone content , ie., with increase in molar ratio in the x axis, the wave number of OH stretch increases, so OH bond length should decrease. Please clarify this part.
Lines 185-186; " The Raman shift of the OH bond is about 8 times of C=O..". As C=O is a double bond , whereas O-H is a single bond, it is generally easier to change the OH bond length compared to C=O bond length. So, to compare the change in wave number , I think a relative percentage change , and also correlation of change in OH /C=O bond length with corresponding change in wave number in Raman spectra , would be of importance.
Author Response
Dear Reviewer:
Thank you for your comments concerning our manuscript (ID: applsci-431965). Thank you very much for giving us the opportunity to modify the paper. Those comments are all valuable and very helpful for revising and improving our paper, as well as the important guiding significance to our researches. We have studied comments carefully and have made correction which we hope meet with approval. Revised portion are marked in blue in the paper. The main corrections in the paper and the responds are as flowing:
Point 1: The paper on " Investigation on surface tension properties of binary solution (acetone-water) based on hydrogen bonding by Raman spectroscopic " , is an important contribution. However, the English has to be improved. Eg., inclusion of either " Raman spectroscopy " or "Raman spectroscopic methods " in the title would be more suitable. There are a lot of places where English has to be improved.
√Dear editor and reviewers,
We have made a substantial revision of the English language in the text, and the modified parts are marked in blue.
Point 2: Some repetition of texts are present, eg., lines 128-131.
√Dear editor and reviewers,
We have made changes to the repeated expressions in the text.
lines125-131.
Point 3: Figure 2,3 or 6 captions contain "acetonitrile ". But this study is based on water and acetone mixtures. Please correct it.
√Dear editor and reviewers,
We have corrected the errors in the figure2, 3, 6 captions.
Point 4: However, in the discussion section, lines 164-166, the authors have argued that the combination of acetone and water reduces the polarity of water and thus OH bond gets elongated and there is a blue shift in Raman spectra. But this is contradictory to Figure 5. As I understood, Figure 5 shows that with increase in acetone content, ie., with increase in molar ratio in the x axis, the wave number of OH stretch increases, so OH bond length should decrease. Please clarify this part.
√Dear editor and reviewers,
Previous analysis is incorrect, and there is a red shift in Raman spectra about OH bond. The specific analysis modified in manuscript.
Line 161-168.
Point 5: Lines 185-186; " The Raman shift of the OH bond is about 8 times of C=O…". As C=O is a double bond, whereas O-H is a single bond, it is generally easier to change the OH bond length compared to C=O bond length. So, to compare the change in wave number, I think a relative percentage change, and also correlation of change in OH /C=O bond length with corresponding change in wave number in Raman spectra, would be of importance.
√Dear editor and reviewers,
The change in the length of OH bond and C=O bond is indeed related to the change in wave number and are worthy of further study. We quoted reference to discuss the change in the length of C=O bond. Specific analysis can be found in the text.
Line 190-197.
We are look forward to hearing from you.
Best regards,
Shunli Ouyang
Corresponding authors
